# Cerebral blood flow autoregulation assessment by correlation analysis between mean arterial blood pressure and transcranial doppler sonography or near infrared spectroscopy is different: A pilot study

**Marcus Thudium**[1]*, **Stefan Moestl**[2], **Fabian Hoffmann**[2,3], **Alex Hoff**[2],
**Evgeniya Kornilov**[4], **Karsten Heusser**[2], **Jens Tank**[2], **Martin Soehle**[1]

**1** Department of Anesthesiology and Intensive Care Medicine, University Hospital Bonn, Venusberg Campus 1, Bonn, Germany, **2** Institute of Aerospace Medicine, German Aerospace Center, Linder Hoehe, Cologne, Germany, **3** Department of Cardiology, University Hospital Cologne, Cologne, Germany, **4** Department of Neurobiology, Weizmann Institute of Science, Rehovot, Israel

* mthu@uni-bonn.de

## Abstract

### Purpose

Recently, cerebral autoregulation indices based on moving correlation indices between mean arterial pressure (MAP) and cerebral oximetry (NIRS, ORx) or transcranial Doppler (TCD)-derived middle cerebral artery flow velocity (Mx) have been introduced to clinical practice. In a pilot study, we aimed to evaluate the validity of these indices using incremental lower body negative pressure (LBNP) until presyncope representing beginning cerebral hypoperfusion as well as lower body positive pressure (LBPP) with added mild hypoxia to induce cerebral hyperperfusion in healthy subjects.

### Methods

Five male subjects received continuous hemodynamic, TCD and NIRS monitoring. Decreasing levels of LBNP were applied in 5-minute steps until subjects reached presyncope. Increasing levels of LBPP were applied stepwise up to 20 or 25 mmHg. Normobaric hypoxia was added until an oxygen saturation of 84% was reached. This was continued for 10 minutes. ORx and Mx indices were calculated using previously described methods.

### Results

Both Indices showed an increase > 0.3 indicating impaired cerebral autoregulation during presyncope. However, there was no significant difference in Mx at presyncope compared to baseline (p = 0.168). Mean arterial pressure and cardiac output decreased only in presyncope, while stroke volume was decreased at the last pressure level. Neither Mx nor ORx showed significant changes during LBPP or hypoxia. Agreement between Mx and ORx was poor during the LBNP and LBPP experiments ($R^2 = 0.001$, p = 0.3339).

**Data Availability Statement:** All relevant data are within the paper and its Supporting Information files.

**Funding:** MT received funding by the Commission for Clinical Studies of the University of Bonn (2018-FKS-04). www.ukbonn.de/studienzentrum-bonn/foerderinstrumente The funders had no role in study design, data collection and analysis, decision to publish, or preparation of the manuscript.

## Conclusion

Mx and ORx represent impaired cerebral autoregulation, but in Mx this may not be distinguished sufficiently from baseline. LBPP and hypoxia are insufficient to reach the upper limit of cerebral autoregulation as indicated by Mx and ORx.

## Introduction

Cerebral autoregulation or the ability to maintain a constant blood flow to the brain during conditions of changing systemic blood pressure is common knowledge since decades [1]. In the clinical setting, both cerebral hyperperfusion and hypoperfusion can have detrimental effects [2, 3]. Absence of cerebral autoregulation was reported to be associated with adverse outcomes in critical care patients [4, 5]. Recently, cerebral autoregulation measurements have been introduced in the perioperative setting, usually based on near infrared spectroscopy (NIRS) as a surrogate parameter for cerebral blood flow [6]. These measurements are based on the assumption that on the so-called Lassen's curve, cerebral blood flow is kept constant over a broad range of mean arterial blood pressure, while above and below this steady-state, cerebral blood flow follows blood pressure changes passively [1, 7]. However, the validity of this model and therefore its applicability in a clinical context is still unclear [8]. Therefore, currently there is no reference method to determine the status of cerebral autoregulation. The aim of the Limits Of Cerebral AutoRegulation (LOCAR) pilot study was to provide an experimental model for cerebral hypoperfusion and hyperperfusion in five healthy volunteers using incremental lower body negative pressure (LBNP) until presyncope and lower body positive pressure (LBPP) combined with mild hypoxia. Thus, it was the objective to produce a state of impaired cerebral autoregulation, below the lower autoregulation limit with LBNP and above the upper autoregulation limit with LBPP and hypoxia [9]. We intended to investigate whether impaired autoregulation is represented by transcranial doppler (TCD) or NIRS-derived autoregulation indices and how this is reflected by global hemodynamic parameters. We hypothesized that autoregulation indices adequately represent impaired autoregulation when exceeding or undercutting the autoregulation thresholds in presyncope and in increased preload and/or increased cardiac output during hypoxia. We further intended to examine the agreement between NIRS and TCD-based autoregulation measurements.

## Materials and methods

This study was approved by the Ethics Committee of the North Rhine Chamber of Physicians (No 2018246). 5 healthy, male volunteers between age 22 and 30 (mean 24.4 ± 2.6 years) were recruited. Mean Height of the subjects was 182.8 ± 3.4 cm, weight 84.8 ± 6.7 kg, BMI 25.4 ± 1.7. Individual biometric data of subjects are shown in S1 Table. None of the subjects had a history of arterial hypertension or antihypertensive medication. After obtaining written informed consent, all subjects underwent standardized medical screening. Although the LBPP experiment was performed before LBNP, the order of presentation is reversed here for reasons of better readability. We did not randomize the protocol since incremental LBNP was applied until presyncope requiring recovery.

### LBNP experiment

Subjects were placed on the experimental table in a supine position and received peripheral venous cannulation. Monitoring was attached consisting of ECG, peripheral oxygen

saturation, continuous blood pressure, stroke volume (SV) and cardiac output (CO) monitoring via fingercuff (Finapres NOVA, Finapres Medical Systems, Enschede, Netherlands), cerebral Near Infrared Spectroscopy (NIRS, NIRO 200NX, Hamamatsu Photonics, Herrsching, Germany), and right temporal transcranial doppler sonography (TCD) via robotic probe (Delica 9UA, Shenzen Medical, Shenzen, China). Their analog output signals were synchronously recorded at 500 Hz by using an analog-digital-converter and then stored on a laptop using the Windaq data acquisition software (Dataq Instruments, Akron, OH, USA). A custom-made airtight chamber was fastened above the lower body of the subject. A Neoprene skirt attached to the chamber and fastened around the waist of the subject prevented air from leaking into the pressure chamber. The lower body chamber was connected to a calibrated air pump that produced a preset pressure which it adjusted automatically according to pressure feedback from a sensor installed inside the chamber. Once all signals were stable, a 10-minute baseline measurement was initiated. Subsequently, lower body pressure was decreased to -15 mmHg for a time of 5 minutes. Pressure was then reduced to -30 mmHg for another 5 minutes. LBNP was further decreased in -5 mmHg steps every 5 minutes. The maximum pressure was maintained until subjects experienced symptoms of presyncope indicated by dizziness, nausea and a sudden loss of blood pressure or cardiac output. At presyncope, LBNP was immediately turned off and the subject was tilted into Trendelenburg position.

## LBPP experiment and hypoxia

For the experiment with positive pressure and hypoxia, subjects were placed on the experimental table as mentioned above. Monitoring was identical to the LBNP experiment. Additionally, subjects received a face mask for in- and expiratory $O_2$ and $etCO_2$ measurements which were acquired with an Innocor gas analyzer (Innovision, Glamsbjerg, Denmark). Once stable signals were achieved on all channels, a baseline measurement of 10 minutes was started. LBPP was then increased in 5 mmHg steps every 5 minutes until the last pressure stage where steady LBPP could be maintained (20 or 25 mmHg), then hypoxia was established with LBPP maintained. Hypoxia was achieved with a gas blender attached to pressurized air and pressurized nitrogen. The gas mixture accumulated in a reservoir bag was subsequently inhaled by the subjects. Hypoxia was reached with an $SpO_2$ of 84%. The experiment was continued for 10 minutes or until abort by the subject. Termination of the experiment was done by turning of the LBPP and detaching the face mask.

## Data acquisition and—analysis

Data acquisition for subsequent off-line beat-to beat analysis was performed with the Windaq system at a sample rate of 500 Hz (Dataq Instruments, Akron, USA). QRS complexes from the ECG, heart rate, systolic-, mean-, and diastolic finger blood pressure were automatically detected and marked based on a custom-written software ("Physiowave") in PV-Wave language (PV-Wave, Visual Numerics, Boulder, USA). Data were manually checked by an experienced observer for wrong or missing values. Further data processing was performed with Matlab (Matlab R2020a, MathWorks Inc. Natick, USA). Cerebral autoregulation indices were calculated using a version of the moving average (Mx) algorithm [3, 6, 10]: A moving Pearson correlation coefficient between mean arterial pressure (MAP) and middle cerebral artery mean flow velocity (MFV, Mx) or MAP and cerebral tissue oxygenation index (TOI, ORx) over 30 ten second averages was used to indicate the state of autoregulation. The resulting Mx and ORx indices are dimensionless values between 0 and 1 representing the degree of linear relationship between MAP and MFV or MAP and TOI, respectively. Increased indices reflect a more pressure passive nature of the underlying TCD or NIRS values. A Pearson coefficient

of 0.3 and above was interpreted as disturbed autoregulation for Mx and ORx based on previous reports [6, 11, 12]. Values for cerebral and global hemodynamic variables as well as autoregulation parameters at the first 10 seconds of the last pressure level in LBNP, 5 seconds before and after presyncope as well as at the last pressure level in LBPP and during hypoxia were compared to baseline measurements (mean values over 10 minutes).

## Statistics

Statistical calculations were performed with SigmaStat 14.0 (Systat Software, San Jose, USA) Comparisons between parameters at baseline, the last pressure level on LBNP/LBPP, and presyncope/hypoxia were done using the One Way Repeated Measured Analysis of Variance (RM ANOVA). A Shapiro-Wilk test was used to test for normal distribution of parameters. Post-hoc-t-tests for multiple pairwise comparisons were performed. P values were automatically corrected for multiple comparisons with Bonferroni correction. Mx and ORx were compared using linear regression analysis. P-values of <0.05 were considered significant.

## Results

A representative example of the complete LBNP and LBPP experiment is shown in Fig 1. Presyncope was reached in all subjects at LBNP levels from -40 mmHg to -70 mmHg. One subject did not tolerate prolonged hypoxia, so the experiment was terminated before reaching the full 10 minutes of measurement time.

During LBNP, presyncope was reached at a MAP of 56 ± 8 mmHg compared to baseline MAP of 95 ± 15 mmHg (-40%, P<0.001) and 93 ± 15 mmHg at the beginning of the last pressure level (-2%, P = 1.00). CO decreased from baseline 5.9 ± 1.2 l/min to 5.4 ± 1.2 l/min (-8%, P = 1.00) at beginning of the last LBNP pressure level and to 3.9 ± 0.79 l/min (-33%, P = 0.023)

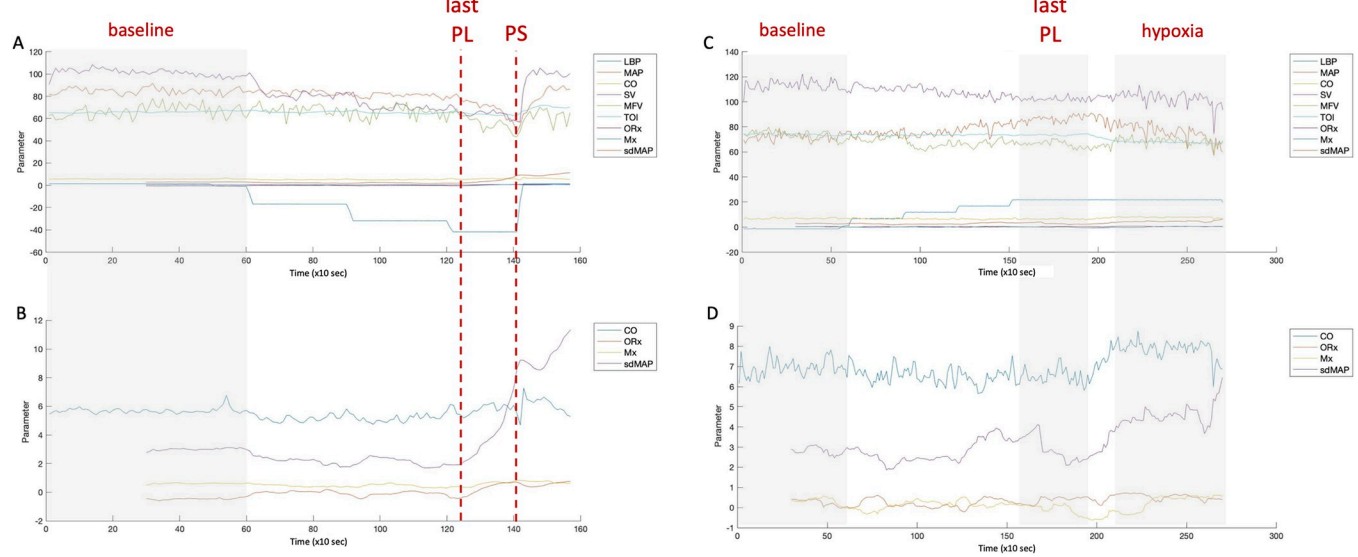

**Fig 1. Representative examples of LBNP and LBPP/hypoxia experiment.** A: All readings from subject 5 during LBNP, where PS marks the onset of presyncope, last PL: last LBNP level, on which presyncope occured B: LBPP/hypoxia, subject 5, enlarged representation of cardiac output and cerebral autoregulation parameters, C: All readings during LBPP, subject 4, last PL marks the last LBPP level before initiation of hypoxia, hypoxia marks steady state hypoxia, D: LBPP/hypoxia, enlarged representation of cardiac and cerebral autoregulation parameters; CO: cardiac output, MAP: mean arterial pressure, Mx: transcranial doppler-derived autoregulation index, LBNP: lower body negative pressure, LBP: lower body pressure, LBPP: lower body positive pressure, ORx: Near-Infrared Spectroscopy derived autoregulation index, sdMAP: moving standard deviation of MAP during 5 minute window, SV: stroke volume, TOI: Tissue Oxygenation Index from Near-Infrared Spectroscopy.

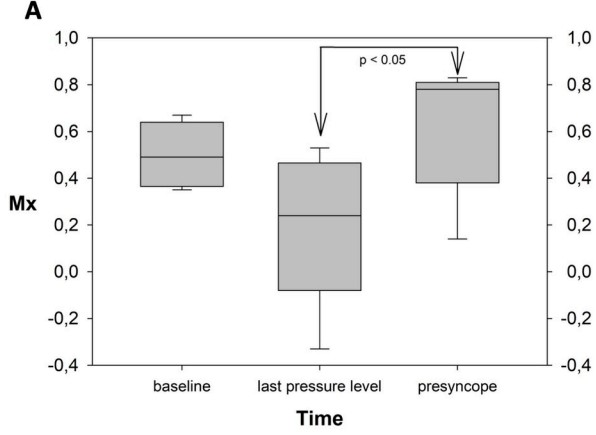

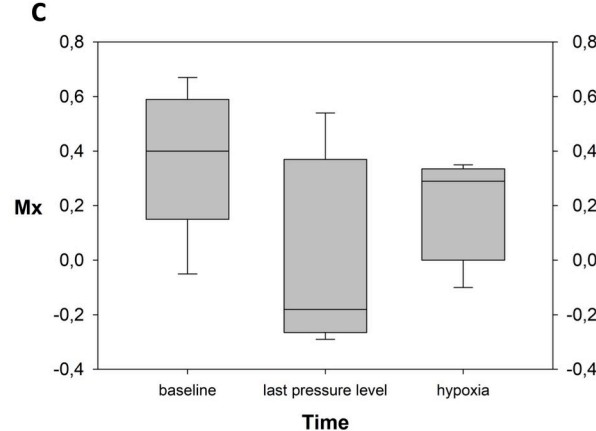

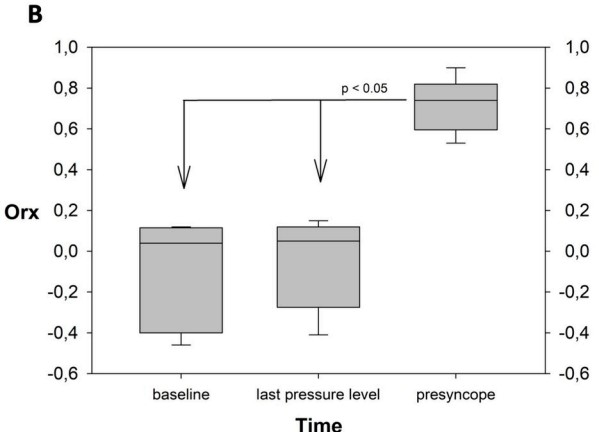

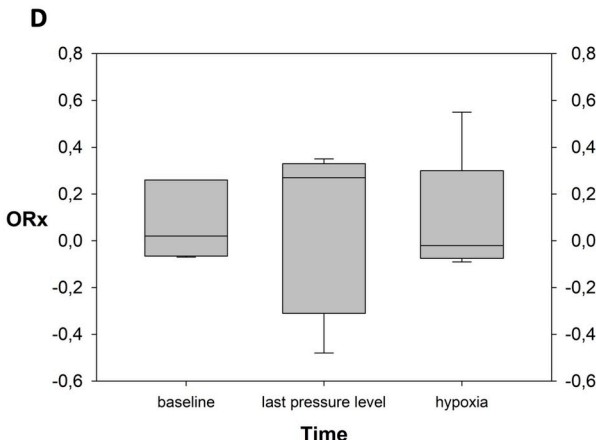

**Fig 2. Statistical evaluation of cerebral autoregulation indices at different time points during the experiment: baseline, last LBNP/LBPP level (LBNP: first 10 seconds), presyncope (5 seconds before, 5 seconds after onset), Mx: transcranial doppler-derived autoregulation index, LBNP: lower body negative pressure, LBPP: lower body positive pressure, ORx: Near-Infrared Spectroscopy derived autoregulation index.**

at presyncope. SV decreased from baseline 90 ± 13 ml to 54 ± 18 ml (-40%, P = 0.023) at beginning of the last LBNP pressure level and to 55 ± 24 ml (-39%, P = 0.026) at presyncope.

MFV decreased from baseline 67 ± 12 cm/s to 59 ± 15 cm/s (-12%, P = 0.093) at the last pressure level (P = 0.093) and decreased to 44 ± 9 cm/s (-35%, P<0.001) in presyncope. TOI changed from baseline 73 ± 4% to 71 ± 4% (-3%, P = 0.248) at the last pressure level and to 69 ± 4% (-6%, P = 0.011) at presyncope. Considering autoregulation parameters, Mx decreased from 0.50 ± 0.13 at baseline to 0.20 ± 0.29 (-60%, P = 0.168) at the beginning of the last pressure level and increased to 0.63 ± 0.26 during presyncope (26%, P = 1.0). However, comparisons showed a significant difference between Mx at the last pressure level and presyncope (P = 0.037). ORx baseline was at -0.11 ± 0.25 and increased to -0.05 ± 0.20 at the last pressure level (52%, P = 1.0) and increased to 0.72 ± 0.12 at presyncope (780%, P<0.001). Differences in ORx were also significant between the last pressure level and presyncope (P<0.001) as also shown in Fig 2.

In the LBPP experiment, baseline MAP was at 92 ± 15 mmHg, increased to 106 ± 19 mmHg (+15% from baseline, P = 0.008) in the last LBPP level and decreased again to 103 ± 22 mmHg (+11%, P = 0.036 from baseline), while CO decreased from baseline 6.6 ± 1.3 l/min to

$6.0 \pm 1.3$ l/min in the last LBPP level (-8%, P = 0.512) and increased with hypoxia to $7.5 \pm 1.5$ l/min (+14% from baseline, P = 0.182). SV changed from $95 \pm 14$ ml to $83 \pm 14$ ml in the last pressure level (-12%, P = 0.020), and was at $86 \pm 17$ ml during hypoxia (-9% from baseline, P = 0.094). Concerning cerebral parameters, MFV decreased from baseline $71 \pm 19$ cm/s to $59 \pm 13$ cm/s at the last LBPP level (-16%, n.s.), and increased again to $70 \pm 13$ cm/s during hypoxia (-1% from baseline, n.s.). TOI was at $75 \pm 4$% during baseline, remained at $75 \pm 3$% in the lase LBPP level, and decreased to $67 \pm 2$% during hypoxia (-11%, P<0.001). Cerebral autoregulation parameters only showed small changes in LBPP and hypoxia. Results of the LBNP and LBPP experiment are summarized in Table 1.

Mx and ORx showed poor agreement in linear regression analysis for the complete experiment without hypoxia as shown in Fig 3 ($R^2$ = 0.001, p = 0.3339).

## Discussion

In this study on five healthy male young adults, we could show that both Mx and ORx represent impaired cerebral autoregulation in LBNP, but show little agreement throughout the LBNP and LBPP experiment. LBPP and hypoxia do not appear to result in impaired cerebral autoregulation.

### LBNP and the limits of cerebral hypoperfusion

The results presented here show that cerebral hypoperfusion begins at a reduction of 35% in MFV of the middle cerebral artery as measured by TCD and a reduction of 5.6% in cerebral tissue oxygen saturation as measured by NIRS. This is in contrast to current accepted limits of the respective methods. A higher volatility of MFV in comparison to cerebral saturation has been described previously on several occasions [13, 14]. Currently, widely accepted thresholds for a reduction in cerebral saturation is a decrease of more than 20% or below 50% absolute value and for TCD a reduction of more than 50% in MFV [15]. It has to be noted that differences in cerebral saturation between NIRS devices from different manufacturers exist [16]. Although these thresholds are mostly based on patients during general anesthesia, there is data on awake patients undergoing carotid endarterectomy, reporting cutoff values of 48% reduction in MFV and 20% reduction in NIRS for development of cerebral ischemia [17]. While one still has to consider the differences to our healthy subjects and the difference in outcome, we suggest a reevaluation of critical limits of both NIRS and TCD. Our results also highlight that in our patients, hypoxia had a larger effect on cerebral saturation than had hypoperfusion, again pointing out that several factors contribute to cerebral oxygen saturation.

In our subjects, both MAP and CO parameters such as SV appear to be involved in the development of cerebral hypoperfusion. While SV was significantly decreased at the last LBNP level this was not reflected by CO. In both MAP and CO, a significant reduction only occurred with presyncope which also reflects cerebral parameters. We interpret this as active regulation mechanisms which compensate decreasing SV before the onset of presyncope which is marked by a decrease in CO which also translates to cerebral parameters. Neumann et al. reported a dependence of cerebral blood flow on CO rather than MAP in hypertensive patients during LBNP [18]. A combined MAP and CO dependence of carotid blood flow has also been shown by Skytioti et al. in surgical patients [19]. The same group have demonstrated an association between CO and carotid blood flow in healthy subjects undergoing LBNP, while MAP remained practically unchanged. Skytioti's LBNP experiment consisted of intermittent LBNP stages of -30 mmHg in contrast to incremental LBNP in our setup that led to presyncope making the results difficult to compare [20].

**Table 1. Procedure related data, LBNP, LBPP and hypoxia.**

| LBNP | | | |
|---|---|---|---|
| **Parameter** | **value** | **change from Baseline, %** | **P** |
| **MAP** | | | |
| • baseline, mmHg | 95.34 ± 15.1 | - | - |
| • last pressure level, mmHg | 93.37 ± 15.25 | -2.07 | 1.00 |
| • presyncope, mmHg | 56.34 ± 8.24 | -40.91 | <0.001 |
| **CO** | | | |
| • baseline, l/min | 5.87 ± 1.2 | - | - |
| • last pressure level, l/min | 5.41 ± 1.16 | -7.8 | 1.00 |
| • presyncope, l/min | 3.89 ± 0.73 | -33.75 | 0.023 |
| **SV** | | | |
| • baseline, ml | 89.56 ± 13.51 | - | - |
| • last pressure level, ml | 53.69 ± 17.73 | -40.05 | 0.023 |
| • presyncope, ml | 54.70 ± 23.87 | -38.92 | 0.026 |
| **MFV** | | | |
| • baseline, cm/s | 67.78 ± 12.01 | - | - |
| • last pressure level, cm/s | 59.78 ± 15.21 | -11.92 | 0.093 |
| • presyncope, cm/s | 44.02 ± 8.77 | -35.15 | <0.001 |
| **TOI** | | | |
| • baseline, % | 73.14 ± 4.16 | - | - |
| • last pressure level, % | 71.29 ± 3.54 | -2.53 | 0.248 |
| • presyncope, % | 69.07 ± 3.54 | -5.56 | 0.011 |
| **Mx** | | | |
| • baseline | 0.50 ± 0.16 | - | - |
| • last pressure level | 0.20 ± 0.29 | -60.13 | 1.000 |
| • presyncope | 0.63 ± 0.26 | 26.47 | 0.168 |
| **Orx** | | | |
| • baseline | -0.11 ± 0.25 | - | - |
| • last pressure level | -0.05 ± 0.20 | 51.82 | 1.000 |
| • presyncope | 0.72 ± 0.12 | 780.18 | <0.001 |
| **LBPP + hypoxia** | | | |
| **MAP** | | | |
| • baseline, mmHg | 92.65 ± 14.90 | - | - |
| • last pressure level, mmHg | 106.56 ± 18.70 | 15.01 | 0.008 |
| • + hypoxia, mmHg | 103.21 ± 22.42 | 11.40 | 0.036 |
| **CO** | | | |
| • baseline, l/min | 6.55 ± 1.27 | - | - |
| • last pressure level (l/min) | 6.04 ± 1.30 | -7.83 | 0.512 |
| • + hypoxia (l/min) | 7.47 ± 1.49 | 14.07 | 0.182 |
| **SV** | | | |
| • baseline, ml | 94.50 ± 13.79 | - | - |
| • last pressure level, ml | 83.17 ± 15.31 | -11.99 | 0.020 |
| + hypoxia, ml | 86.39 ± 17.17 | -8.58 | 0.094 |
| **MFV** | | | |
| • baseline, cm/s | 70.53 ± 18.81 | - | - |
| • last pressure level, cm/s | 59.34 ± 12.88 | -15.87 | n.s. |
| • + hypoxia, cm/s | 69.80 ± 12.84 | -1.03 | n.s. |
| **TOI** | | | |

*(Continued)*

**Table 1.** (Continued)

| LBNP | | | |
|---|---|---|---|
| **Parameter** | **value** | **change from Baseline, %** | **P** |
| • **baseline, %** | 74.50 ± 3.86 | - | - |
| • **last pressure level, %** | 74.65 ± 3.31 | 0.20 | 1.000 |
| • **+ hypoxia, %** | 66.49 ± 1.96 | -10.75 | <0.001 |
| **Mx** | | | |
| • **baseline** | 0.38 ± 0.24 | - | - |
| • **last pressure level** | 0.01 ± 0.32 | -98.31 | n.s. |
| • **+ hypoxia** | 0.19 ± 0.17 | -48.34 | n.s. |
| **ORx** | | | |
| • **baseline** | 0.08 ± 0.15 | - | - |
| • **last pressure level** | 0.06 ± 0.32 | -23.32 | n.s. |
| • **+ hypoxia** | 0.08 ± 0.24 | 3.33 | n.s. |

Data is presented as mean ± standard deviation

While the exact threshold for impaired autoregulation remains unclear and is reported to be at index values of 0.3 to 0.5, a threshold of 0.3 has mostly been used recently [6, 11, 21]. Using this threshold, both Mx and ORx adequately represent impairment in autoregulation during presyncope. However, while there was a large increase in ORx, the difference in Mx compared to baseline was less distinct, since Mx was already elevated above 0.3 at baseline. One has to keep in mind the different measurement modalities of TCD and NIRS, the one representing exclusively arterial flow while the other is a mostly venous signal [13, 14]. Although the meaning of an impaired TCD-derived autoregulation at baseline is unclear at this point, we have to assume that this is a physiological phenomenon in our cohort of healthy subjects. Therefore, these results question the ability of the Mx to discriminate between intact and impaired autoregulation in contrast to the ORx. The use of a higher Mx threshold of 0.5 would not have changed this finding [3]. One may suspect from these results that TCD-derived indices are more sensitive, as we have also suspected previously in a clinical context [14]. Our findings may also confirm the hypothesis that the plateau region on the autoregulation curve is not as wide as previously assumed and may be missed by some measurement modalities, or alternatively may not be present at all [8]. At the beginning of the last pressure level at which presyncope occurred we did not observe significant changes in Mx or ORx, while both were below the 0.3 threshold. It seems that cerebral autoregulation was still intact at this point even with imminent presyncope. We suspect that in our collective of young healthy subjects, regulatory mechanisms may be active until the final collapse of hemodynamics in presyncope which is only then reflected by increased Mx and ORx. We also suspect differences to older, morbid patients or patients during general anesthesia. However, more data is needed to investigate the relationship between autoregulation indices, age and medical history.

## LBPP/hypoxia

While LBNP has been studied extensively, the same does not apply to LBPP. We observed a decrease in MFV while TOI remained unchanged at the last pressure level of LBPP. CO and SV decreased while there was a substantial increase in MAP. These hemodynamic findings are in line with previous head-down tilt studies representing the same physiological mechanism [22]. Perry et al. reported unchanged cardiac output, but increased MCA flow velocity in moderate LBPP of 20 mmHg, but unchanged MFV in high LBPP of 40 mmHg which is explained

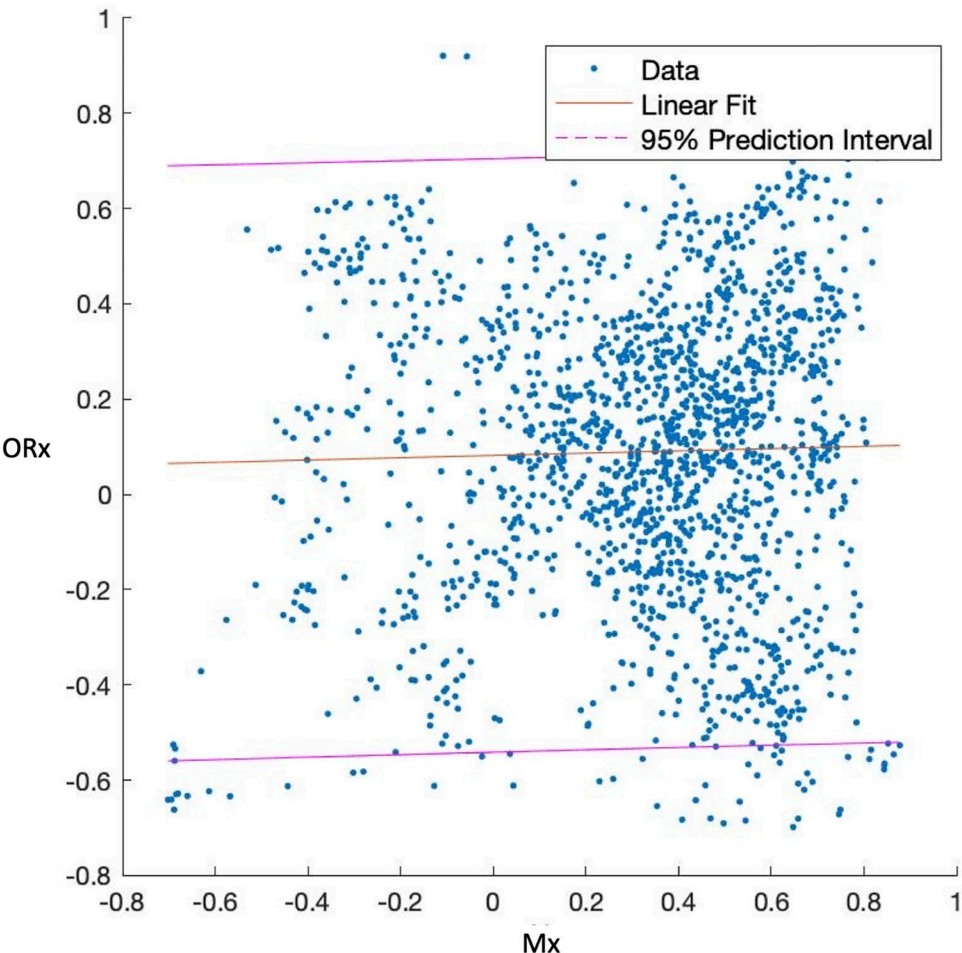

**Fig 3. Linear regression analysis of agreement between Mx and ORx during LBNP and LBPP experiment, all subjects; hypoxia was excluded due to its effect on cerebral saturation; Mx: transcranial Doppler-derived autoregulation index, LBNP: lower body negative pressure, LBPP: lower body positive pressure, ORx: Near-Infrared Spectroscopy derived autoregulation index.**

by alterations in baroreflex sensitivity [23]. Despite lower LBPP levels in our subjects, our findings are still in line with Perry et al, although we would explain the decrease in MFV rather mechanistically with the higher postcapillary resistance once a level of stressed volume of the venous system is reached. Mx and ORx calculations did not indicate impaired autoregulation during LBPP, nor was this the case for hypoxia, despite a significant increase in MAP and increased CO during hypoxia. This is somewhat surprising since impaired autoregulation has been reported to occur in acute hypoxia on several occasions [9, 24–26]. Whether the hypoxia stimulus was insufficient to cause impaired autoregulation or indices did not adequately show impaired autoregulation is unclear. TCD values appeared unaffected by LBPP or hypoxia which may indicate cerebral hemodynamic changes in hypoxia were not sufficient to cause autoregulation impairment. However, hypoxia acts as a powerful vasodilatator which has been shown to alter TCD readings [27, 28]. Hypocapnia may also play a role through hyperventilation [29]. TOI naturally showed a substantial decrease during hypoxia. This is important for clinical practice since changes in oxygen saturation may be more pronounced in NIRS than changes in cerebral circulation. In NIRS, cerebral malperfusion may thus be masked by changes in oxygen settings. Therefore, with the results presented here we cannot confirm the

use of Mx or ORx for the determination of the upper autoregulation limit. For future studies we suggest different methods to reach the upper autoregulation limit, such as exhaustive exercise or a combination of exercise and LBPP [30, 31].

## Agreement between autoregulation indices

Both Mx and ORx represent autoregulation impairment in presyncope using a threshold of 0.3 to 0.5, although the impairment is more distinct in ORx. As shown in Fig 3, Mx and ORx show poor agreement throughout the LBNP and LBPP experiment, excluding hypoxia. It appears that Lassen's curve and its relationship to global hemodynamic parameters depends on the location and modality of measurement. What can be stated at this point is that Mx and ORx are not interchangeable and measurements from different modalities may only be compared with caution. This is in contrast to a report by Steiner et al. on intensive care patients with sepsis, where an almost perfect agreement between the two methods has been shown [32]. The difference in our findings may be caused by awake subjects with spontaneous breathing unlike Steiner's patients or the underlying disease which may be associated in compromised regulation mechanisms of the cerebral vasculature. Cortical activation in awake subjects may further influence results, while cortical activity should remain relatively unchanged in sedated patients. While the exact cause of this disparity remains unclear at this point, it can be stated that Mx and ORx methods yield different results in awake subjects. However, further validation studies are necessary to confirm these results.

## Limitations

Several limitations are associated with this study. First, this is a pilot study with few subjects, therefore it is difficult to generalize the presented results. Second, NIRS and TCD both represent surrogate parameters for cerebral blood flow which both have inherent technical limitations affecting measurement precision and accuracy such as an unknown vascular diameter and dependence on the insonification angle in TCD and for NIRS the possibility of extracranial contamination of the signal. The same is the case for non-invasive CO and MAP measurements which may present inaccuracies, especially in the more extreme ranges.

## Conclusions

In summary, in our pilot study on healthy male subjects we could show that while NIRS and TCD-derived autoregulation indices Mx and ORx both indicated presyncope in LBNP, they showed poor agreement throughout the LBNP and LBPP experiment. During presyncope, changes in Mx were less distinct than in ORx. Thus, our results support the use of ORx to indicate impaired autoregulation while the same remains questionable with Mx. During LBPP alone or in the combination with hypoxia the upper limit of cerebral autoregulation could not be observed with Mx or ORx in our subjects. We suggest that other methods be used to reach the upper limit of autoregulation. In awake subjects, direct comparisons between Mx and ORx should only be done with care. These results should be confirmed in a larger study.

## Supporting information

**S1 Table. Individual biometric data of patients.**
(XLSX)

**S2 Table. Beat-to beat data of all patients, all experiments.**
(XLSX)

## Author Contributions

**Conceptualization:** Marcus Thudium, Stefan Moestl, Fabian Hoffmann, Alex Hoff, Evgeniya Kornilov, Karsten Heusser, Jens Tank.

**Data curation:** Marcus Thudium.

**Formal analysis:** Marcus Thudium, Martin Soehle.

**Funding acquisition:** Marcus Thudium.

**Investigation:** Marcus Thudium, Stefan Moestl, Fabian Hoffmann, Alex Hoff.

**Methodology:** Evgeniya Kornilov, Jens Tank.

**Project administration:** Marcus Thudium.

**Supervision:** Jens Tank, Martin Soehle.

**Validation:** Karsten Heusser, Martin Soehle.

**Visualization:** Martin Soehle.

**Writing – original draft:** Marcus Thudium, Evgeniya Kornilov, Martin Soehle.

**Writing – review & editing:** Marcus Thudium, Stefan Moestl, Fabian Hoffmann, Alex Hoff, Evgeniya Kornilov, Karsten Heusser, Jens Tank, Martin Soehle.

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
