## [Decision Letter · Decision Letter 0]

27 Mar 2023

PONE-D-22-35647

Cerebral Blood Flow Autoregulation Assessment by Correlation Analysis Between Mean Arterial Blood Pressure and Transcranial Doppler Sonography or Near Infrared Spectroscopy is Different

PLOS ONE

Dear Dr. Thudium,

Thank you for submitting your manuscript to PLOS ONE. After careful consideration, we feel that it has merit but does not fully meet PLOS ONE’s publication criteria as it currently stands. Therefore, we invite you to submit a revised version of the manuscript that addresses the points raised during the review process.

Please address all 3 reviewers' comments in detail and see my additional editor comments!

We look forward to receiving your revised manuscript.

Kind regards,

Stephan Meckel, MD, PhD

Academic Editor

PLOS ONE

Journal Requirements:

2. In the Methods section, please provide additional details regarding participant consent. In the ethics statement in the Methods and online submission information, please ensure that you have specified what type you obtained (for instance, written or verbal, and if verbal, how it was documented and witnessed). If your study included minors, state whether you obtained consent from parents or guardians. If the need for consent was waived by the ethics committee, please include this information.

Additional Editor Comments:

This study does not fulfill the criteria of a complete validation with only 5 subjects tested and severe methodological shortcomings. Though some reviewers found aspects that may merit publication.

Please address all shortcomings and make clear that the study is termed pilot study only in title, abstract and main manuscript. Leave out the term validation and indicate that further validation studies are necessary!

Please address all comments by the three reviewers and add a statistical correction for multiple testing!

Reviewers' comments:

Reviewer's Responses to Questions

**Comments to the Author**

1. Is the manuscript technically sound, and do the data support the conclusions?

Reviewer #1: Partly

Reviewer #2: Yes

Reviewer #3: Yes

2. Has the statistical analysis been performed appropriately and rigorously? 

Reviewer #1: Yes

Reviewer #2: Yes

Reviewer #3: No

3. Have the authors made all data underlying the findings in their manuscript fully available?

Reviewer #1: No

Reviewer #2: Yes

Reviewer #3: Yes

4. Is the manuscript presented in an intelligible fashion and written in standard English?

Reviewer #1: Yes

Reviewer #2: Yes

Reviewer #3: Yes

5. Review Comments to the Author

Reviewer #1: The present study was to test the validity of NIRS and transcranial Doppler sonography (TCD)-derived autoregulation indices by exceeding the limits of cerebral autoregulation.

However, I do not understand how the authors confirmed the validity of these methods. The authors only compared cerebral autoregulation indexes determined by NIRS and TCD during hyper- and hypotensive conditions. The authors should provide a method to test the validity of these indexes (for example, indicating the gold standard method to identify cerebral autoregulation). Also, the rationale and research impact of the present study is unclear. Both indexes have been already established, thus the authors should provide the present issue regarding these indexes (why the authors needed to test the validity of these methods).

Moreover, the conclusion is not matched their aim. The authors should conclude regarding the validity of these methods. In addition, the results of cerebral autoregulation response to hypo- and hypertensive stimulation are not new. The authors need to demonstrate some importance to these results in the limited issue in this research area with the adequate rationale of this study.

Reviewer #2: The authors are to be congratulated on their article evaluating cerebral autoregulation indices based on moving correlation indices between mean arterial pressure (MAP) and cerebral oximetry (NIRS, ORx) or transcranial Doppler (TCD)-derived middle cerebral artery flow velocity (Mx) evaluating their validity using incremental lower body negative pressure (LBNP) until presyncope in healthy subjects.

Grammar is acceptable in English as a first language with adequate figures and tables.

Methods: This is a small prospective pilot that is unblinded small using Five male subjects receiving continuous hemodynamic, TCD and NIRS monitoring. The precision and accuracy for the final measure have not been clearly established. It is not clear how cardiac parameters such as inidvidual EF, VO2 max, and essential hypertension are related to individual participant responses. Additional data on body baitus and any metric outlining cardiovascular fitness has not been included. Additionally no woman were included in the study. The units and confidence intervals of the two Mx and ORx should be listed or illustrated as derived figures.

Investigation involved this protocol: “The investigators Decreasing levels of LBNP were applied in 5-minute steps until subjects reached presyncope. Increasing levels of LBPP were applied stepwise up to 20 or 25 mmHg. Normobaric hypoxia was added until an oxygen saturation of 84% was reached. This was continued for 10 minutes. ORx and Mx indices were calculated using previously described methods.”

Results: Both Indices showed an increase > 0.3 indicating impaired cerebral autoregulation during presyncope. However, there was no significant difference in Mx at presyncope compared to baseline (p=0.168). Mean arterial pressure and cardiac output decreased only in presyncope, while stroke volume was decreased at the last pressure level. Neither Mx nor ORx showed significant changes during LBPP or hypoxia. Agreement between Mx and ORx was poor during the LBNP and LBPP experiments (R2=0.001, p=0.3339).

The conclusions are correct as stated and supported by the data: “Conclusion: Mx and ORx represent impaired cerebral autoregulation, but in Mx this may not be distinguished sufficiently from baseline. LBPP and hypoxia are insufficient to reach the upper limit of cerebral autoregulation as indicated by Mx and ORx.”

1) This article is an exiting new pilot study that is addressing the fundamentals of how these indices work. It is wel suited for the unconventional and probing science that PLOS was designed for.

2) The authors have outlined an interesting and new experimental protocol.

3) The authors have provided an outline for future studies.

4) There are issues pertaining to precison and accuracy of the instruments involved both technical and epidemiological. Many physiological measures are derived from calculations and the precision and accuracy of thee are not clear to me.

Reviewer #3: The manuscript is very clear. The only concern is that with the many comparisons that are made no correction for multiple comparisons is applied. After correction some of the results that are now presented as significantly different might not be significantly different anymore, which alters the discussion section.

6. PLOS authors have the option to publish the peer review history of their article (what does this mean?). If published, this will include your full peer review and any attached files.

Reviewer #1: No

Reviewer #2: **Yes: **Paul Nyquist MD/MPH, FCCM, FANA, FAAN, FAHA

Reviewer #3: No

---

## [Author Response · Author response to Decision Letter 0]

12 May 2023

We would like to thank the reviewers for their valuable input and constructive feedback, which greatly aided in the refinement of our manuscript. We are confident that we have successfully addressed all the points raised, and we present below a detailed, point-by-point response to both the reviewers’ comments and the editor’s comments

Additional Editor Comments:

This study does not fulfill the criteria of a complete validation with only 5 subjects tested and severe methodological shortcomings. Though some reviewers found aspects that may merit publication.

Please address all shortcomings and make clear that the study is termed pilot study only in title, abstract and main manuscript. Leave out the term validation and indicate that further validation studies are necessary!

The term “pilot study” has been added. The term “validation” has been avoided and we mentioned that further validation is necessary.

Please address all comments by the three reviewers and add a statistical correction for multiple testing!

All comments have been addressed. A correction for multiple measurements is now included. 

Reviewer #1: The present study was to test the validity of NIRS and transcranial Doppler sonography (TCD)-derived autoregulation indices by exceeding the limits of cerebral autoregulation.

However, I do not understand how the authors confirmed the validity of these methods. The authors only compared cerebral autoregulation indexes determined by NIRS and TCD during hyper- and hypotensive conditions. The authors should provide a method to test the validity of these indexes (for example, indicating the gold standard method to identify cerebral autoregulation). Also, the rationale and research impact of the present study is unclear. Both indexes have been already established, thus the authors should provide the present issue regarding these indexes (why the authors needed to test the validity of these methods).

We thank the reviewer for this important comment and we agree that this has not been clearly stated in the manuscript. As much as we would like to present a gold standard for cerebral autoregulation assessment, currently there is none (see also Klein et al. Critical Care 2019, 23:160). Therefore, the experiment itself had to provide the gold standard: With the onset of presyncope, cerebral hypoperfusion occurs, so by definition alone this is beyond the lower limit of cerebral autoregulation.

However, determining the upper limit is more complicated. In our study, we applied increased preload with LBPP and induced mild hypoxia to shift hemodynamics to the right side of the autoregulation curve. While we observed hemodynamic changes, we could not discern impaired cerebral autoregulation using LBPP and hypoxia.

We also agree with the reviewer that Mx has been the most and longest used of the methods and is now well established together with its NIRS counterpart ORx. However, the validity of both methods is far from safe. Only recently, the validity of Mx has been described as heterogenous. Additionally, recent discoveries that the “plateau” of Lassen’s curve may not be as flat or as wide as originally suggested imply that the methods may not suffice to distinguish between intact and impaired cerebral autoregulation. Therefore, we saw an indication to evaluate both Mx and ORx and their suitability for a wider application and for larger studies in this context.

To address this, we revised the introduction section and hope that the background is more understandable.

Moreover, the conclusion is not matched their aim. The authors should conclude regarding the validity of these methods. In addition, the results of cerebral autoregulation response to hypo- and hypertensive stimulation are not new. The authors need to demonstrate some importance to these results in the limited issue in this research area with the adequate rationale of this study.

We thank the reviewer for this comment. The conclusion has been changed accordingly. In addition, more emphasis has been put on validity in the discussion section as follows:

LBNP and the limits of cerebral hypoperfusion: We added a sentence stating “Therefore, these results question the ability of the Mx to discriminate between intact and impaired autoregulation in contrast to the ORx.”

LBPP/Hypoxia: We added “Therefore, with the results presented here we cannot confirm the use of Mx or ORx for the determination of the upper autoregulation limit.”

Agreement between Autoregulation Indices: We changed the last sentences which now read “While the exact cause of this disparity remains unclear at this point, it can be stated that Mx and ORx methods yield different results in awake subjects thus questioning the validity of these methods. However, further validation studies are necessary to confirm these results.”

While we agree that a variety of LBNP experiments have been presented, this is not the case for LBPP. Additionally, the combination of TCD and NIRS is rarely used and the discrepancy between Mx and ORx shown here has not been observed previously. Since both indices are certainly the most widely used, users of these methods should be aware that interpretation has to be done with care. Due to a small sample size, these results will have to be confirmed in a larger cohort. We also mentioned this in the conclusion.

Reviewer #2: The authors are to be congratulated on their article evaluating cerebral autoregulation indices based on moving correlation indices between mean arterial pressure (MAP) and cerebral oximetry (NIRS, ORx) or transcranial Doppler (TCD)-derived middle cerebral artery flow velocity (Mx) evaluating their validity using incremental lower body negative pressure (LBNP) until presyncope in healthy subjects.

Grammar is acceptable in English as a first language with adequate figures and tables.

Methods: This is a small prospective pilot that is unblinded small using Five male subjects receiving continuous hemodynamic, TCD and NIRS monitoring. The precision and accuracy for the final measure have not been clearly established. It is not clear how cardiac parameters such as inidvidual EF, VO2 max, and essential hypertension are related to individual participant responses. Additional data on body baitus and any metric outlining cardiovascular fitness has not been included. Additionally, no woman were included in the study. The units and confidence intervals of the two Mx and ORx should be listed or illustrated as derived figures.

We aimed to keep participants as homogenous as possible to obtain comparable results in this small sample size. Therefore, only male subjects between age 20 and 30 were included who were in perfect health. Including women would have made interpretation difficult at best. Furthermore, subjects underwent standardized medical examination consisting of a detailed medical history, blood count, pulmonary function test and ecg. Only patients with normal values could be included in the study. Biometric parameters of the subjects have been added in S1 Table. 

We apologize since autoregulation indices have not been explained sufficiently. Since both Mx and ORx are based on a Pearson correlation coefficient, these are dimensionless indices between 0 and 1 representing the pressure-passive nature of cerebral perfusion parameters. We added an explanation to the Methods section. Mean ± standard deviation of values are shown in table 1, these should be the correct measures. Additional figures can be provided as the reviewer deems necessary. Precision or accuracy of ORx/Mx cannot be indicated, especially since there is no reference method. The underlying measurement of TCD and NIRS have been extensively validated. However, accuracy of NIRS is still not entirely clear and depends on the device used (see also Anesth Analg. 2013 Oct;117(4):813-823). 

Investigation involved this protocol: “The investigators Decreasing levels of LBNP were applied in 5-minute steps until subjects reached presyncope. Increasing levels of LBPP were applied stepwise up to 20 or 25 mmHg. Normobaric hypoxia was added until an oxygen saturation of 84% was reached. This was continued for 10 minutes. ORx and Mx indices were calculated using previously described methods.”

Results: Both Indices showed an increase > 0.3 indicating impaired cerebral autoregulation during presyncope. However, there was no significant difference in Mx at presyncope compared to baseline (p=0.168). Mean arterial pressure and cardiac output decreased only in presyncope, while stroke volume was decreased at the last pressure level. Neither Mx nor ORx showed significant changes during LBPP or hypoxia. Agreement between Mx and ORx was poor during the LBNP and LBPP experiments (R2=0.001, p=0.3339).

The conclusions are correct as stated and supported by the data: “Conclusion: Mx and ORx represent impaired cerebral autoregulation, but in Mx this may not be distinguished sufficiently from baseline. LBPP and hypoxia are insufficient to reach the upper limit of cerebral autoregulation as indicated by Mx and ORx.”

1) This article is an exiting new pilot study that is addressing the fundamentals of how these indices work. It is well suited for the unconventional and probing science that PLOS was designed for.

2) The authors have outlined an interesting and new experimental protocol.

3) The authors have provided an outline for future studies.

4) There are issues pertaining to precison and accuracy of the instruments involved both technical and epidemiological. Many physiological measures are derived from calculations and the precision and accuracy of thee are not clear to me.

We agree with the reviewer that there are issues concerning the precision and accuracy of both TCD and NIRS. These are inherent in the methods and cannot be addressed easily such as unknown vessel diameter in TCD and extracranial contamination in NIRS. We added potential inaccuracies to the limitations section.

Reviewer #3: The manuscript is very clear. The only concern is that with the many comparisons that are made no correction for multiple comparisons is applied. After correction some of the results that are now presented as significantly different might not be significantly different anymore, which alters the discussion section.

We thank the reviewer for highlighting this and we apologize since the reviewer is correct that this part has been incomplete. After a significant result in the ANOVA, multiple pairwise comparisons were performed with post-hoc t-tests (Bonferroni t-test). P values were corrected by multiplying by the number of comparisons. We added this to the methods section. Since this is a pilot study, the analysis is of exploratory nature. Therefore, we avoided to overinterpret significant or non-significant values and now avoid the term “significant”. We hope that this is in line with the reviewer’s expectations.

---

## [Decision Letter · Decision Letter 1]

8 Jun 2023

Cerebral Blood Flow Autoregulation Assessment by Correlation Analysis Between Mean Arterial Blood Pressure and Transcranial Doppler Sonography or Near Infrared Spectroscopy is Different: A Pilot Study

PONE-D-22-35647R1

Dear Dr. Thudium,

We’re pleased to inform you that your manuscript has been judged scientifically suitable for publication and will be formally accepted for publication once it meets all outstanding technical requirements.

Kind regards,

Stephan Meckel, MD, PhD

Academic Editor

PLOS ONE

Additional Editor Comments (optional):

Reviewers' comments:

Reviewer's Responses to Questions

**Comments to the Author**

1. If the authors have adequately addressed your comments raised in a previous round of review and you feel that this manuscript is now acceptable for publication, you may indicate that here to bypass the “Comments to the Author” section, enter your conflict of interest statement in the “Confidential to Editor” section, and submit your "Accept" recommendation.

Reviewer #2: All comments have been addressed

Reviewer #3: All comments have been addressed

2. Is the manuscript technically sound, and do the data support the conclusions?

Reviewer #2: Yes

Reviewer #3: Yes

3. Has the statistical analysis been performed appropriately and rigorously? 

Reviewer #2: Yes

Reviewer #3: Yes

4. Have the authors made all data underlying the findings in their manuscript fully available?

Reviewer #2: Yes

Reviewer #3: Yes

5. Is the manuscript presented in an intelligible fashion and written in standard English?

Reviewer #2: Yes

Reviewer #3: Yes

6. Review Comments to the Author

Reviewer #2: They have addressed my concerns. The article is nicely improved. The study is a small methods paper and an important first step.

Reviewer #3: Authors addressed the main point, so there is no further comment to make. There is not further comment

7. PLOS authors have the option to publish the peer review history of their article (what does this mean?). If published, this will include your full peer review and any attached files.

Reviewer #2: No

Reviewer #3: No

---

## [Editor Report · Acceptance letter]

13 Jun 2023

PONE-D-22-35647R1 

Cerebral Blood Flow Autoregulation Assessment by Correlation Analysis Between Mean Arterial Blood Pressure and Transcranial Doppler Sonography or Near Infrared Spectroscopy is Different: A Pilot Study 

Dear Dr. Thudium:

I'm pleased to inform you that your manuscript has been deemed suitable for publication in PLOS ONE. Congratulations! Your manuscript is now with our production department. 

Kind regards, 

on behalf of

Prof. Dr. Stephan Meckel 

Academic Editor

PLOS ONE